# Some Physiological Responses of Native Sheep Breeds to Environmental Conditions during Grazing in Natura 2000 Habitats

Monika Greguła-Kania [1], Paulina Nazar [1,*], Mariusz Kulik [2], Krzysztof Patkowski [1], Agata Hahaj-Siembida [3] and Andrzej Junkuszew [1]

1   Department of Animal Breeding and Agricultural Consulting, University of Life Sciences in Lublin, Akademicka 13, 20-950 Lublin, Poland
2   Department of Grassland and Landscape Forming, University of Life Sciences in Lublin, Akademicka 15, 20-950 Lublin, Poland
3   Department of Preclinical Veterinary Sciences, University of Life Sciences in Lublin, Akademicka 15, 20-950 Lublin, Poland
*   Correspondence: paulina.dudko@up.lublin.pl

**Abstract:** The calcareous xerothermic grasslands of the Festuco-Brometea class are among the most endangered Natura 2000 sites in Europe. The greatest threats include a lack of grazing, secondary succession, and invasive alien and expansive native species. The abandonment of such grasslands is caused by the low nutritional value of biomass, as well as by difficult habitat conditions. The aim of this study was to assess the physiological responses of native sheep breeds to hard environmental conditions when grazing in Natura 2000 habitats and to confirm the hypothesis that native sheep of the Świniarka and Uhruska breeds can be used to protect valuable natural habitats. The analysed grasslands were characterised by very difficult climatic and edaphic conditions and a low fodder value for sheep. Grazing in environmentally valuable areas, sometimes poor in fodder, did not adversely affect the welfare of grazed sheep. In both the Uhruska and Świniarka breeds, there were no indicators for heat or nutritional stress, but physiological changes were observed in the animals' adaptation to new environmental conditions. Despite poor fodder positions, Świniarka sheep improved their condition and body weight. Fluctuations in indicators showing an adaptation process were also visible.

**Keywords:** Świniarka breed; Uhruska breed; Festuco-Brometea class; calcareous xerothermic grasslands

## 1. Introduction

The calcareous xerothermic grasslands of the Festuco-Brometea class are among the most endangered Natura 2000 sites in Europe. These habitats are a sanctuary for many rare and endangered species of fauna and flora. Dry grasslands belong to seminatural ecosystems, which means that they require extensive management. Unfortunately, in recent decades, many dry grasslands have been abandoned, which has resulted in habitat changes [1]. The greatest threats to xerothermic grasslands include a lack of grazing, secondary succession, invasive alien and expansive native species, and dead organic matter accumulation, which lead to successive changes in the habitat [1–4]. The abandonment of such grasslands is caused by the low nutritional value of biomass, as well as by difficult habitat conditions. Calcareous xerothermic grasslands are often located on steep slopes or in places with difficult access. In view of the decreasing number of farm animals, including sheep, the protection of xerothermic grasslands is a difficult challenge requiring precise grazing planning, especially since not all farm animals can be assured of welfare in such difficult conditions. The nutritional requirements of high-production ruminants are greater than those of native breeds. In such habitats, native breeds, mainly sheep, are best suited to local environmental conditions in terms of diet and maintenance [5–8].

The definition of animal welfare is the extent to which an animal is coping with its environment and the expression of positive emotional states [9,10]. Animal care should be assessed by a variety of measures: physiological and blood morphological parameters as well as behaviour observations. The task of the breeder is to create conditions that will ensure proper living comfort for the animals. Threats related to animal welfare are primarily associated with intensive animal production. However, during extensive livestock production technologies, there may also be factors causing increased animal stress, for example: (a) environmental factors, such as extreme climatic conditions or poor water quality or availability; (b) psychological factors, such as inappropriate human–animal interactions; (c) physiological and nutritional stress, such as lack of a properly balanced diet and malnutrition; (d) misbehaviour stress, such as transport stress and animal movement; and (e) stress related to pathological conditions, such as lameness and diarrhoea [11,12]. In accordance with the recommendations of in situ protection, local breeds of livestock should be kept in an extensive system using natural conditions, e.g., pastures, and those in environmentally valuable areas [13]. Therefore, it is necessary to ensure that the farming system is supported by research on the impact of the environment on animal welfare.

The aim of this study was (1) to assess the physiological responses of native sheep breeds living in hard environmental conditions when grazing in Natura 2000 habitats based on physiological indicators and (2) to confirm the hypothesis that native sheep of the Świniarka and Uhruska breeds can be used to protect valuable natural habitats.

## 2. Materials and Methods

### 2.1. Study Area

This research was carried out in eastern Poland at three Natura 2000 sites (Table 1). Xerothermic grasslands in Gródek are located on a slope and represent a small part of the PLH 060035 Natura 2000 site (1556.11 ha area), while Kąty II is part of the PLH 060010 (23.98 ha area). Stawska Góra, located in Staw village, is both a flora reserve (4.9 ha of area) and a Natura 2000 site and constitutes a unique sanctuary for many rare, relict Natura 2000 plant species, including *Carlina acanthifolia* subsp. *utzka* [14].

**Table 1.** Natura 2000 sites and phytosociological classification of xerothermic grassland.

| Nature 2000 Site/Place | Vegetation Type | Dominant Species |
|---|---|---|
| Stawska Góra PLH 060018/Staw | All. *Cirsio-Brachypodion pinnati* | *Carlina acanthifolia* subsp. *utzka, Brachypodium pinnatum, Melampyrum arvense* |
| | Cl. *Rhamno-Prunetea* | *Rhamnus catharticus, Cornus sanguinea, Prunus spinose, Viburnum opulus* |
| Zachodniowołyńska Dolina Bugu PLH 060035/Gródek | Ass. *Thalictro-Salvietum pratensis* | *Elymus hispidus* subsp. *hispidus, Salvia pratensis* |
| | community with *Brachypodium pinnatum* | *Brachypodium pinnatum* |
| Kąty PLH 060010/Kąty | Ass. *Inuletum ensifoliae* | *Inula ensifolia, Linum flavum* |
| | initial community | *Anthyllis vulneraria* |
| | community with *Elymus repens* | *Elymus repens, Hieracium piloselloides, Taraxacum officinale, Achillea millefolium* |

### 2.2. Animals

The Local Ethics Committee for Animal Experimentation in Lublin, Poland, (License No 2/2015) approved this study. We enrolled 40 ewes (3–4 years of age) of the native breeds of an area in Poland: Uhruska and Świniarka (20 ewes of each breed). There were no male animals and no lambs. The Uhruska breed is a subvariety of the Lublin and originated from crosses of Leine and Romney rams with Merino ewes in the late 1950s. They are medium-large sheep. Mature rams weigh 100 kg and ewes weigh 65 kg. They are found in the Lublin area of central-eastern Poland [15].

The Świniarka breed is critically endangered. The sheep are small, in primitive type, and short-tailed. Rams weigh 45 kg and have wide-spreading spiral horns. Ewes weigh 30 kg and have short horns or are polled. They are thrifty, disease resistant, and well adapted to poor conditions and feed of low quality [15]. Before and after the grazing period, the animals were kept under identical nutritional and environmental conditions in an indoor-pasture system in a commercial herd located in Bezek, Poland. From the late autumn to spring, sheep are kept in sheepfolds. From the spring to late autumn, animals spend a lot of hours per day on pasture. They spend nights in sheepfolds. The rotational grazing of native sheep breeds was conducted in Gródek, Kąty, and Staw from the beginning of June to mid-October 2018 for approximately 7 weeks in each place in free grazing conditions. During grazing, animals had constant access to water and shade, and during the night they were kept in a shed as a shelter from the rain, wind, and predators. The pasture was fenced by electric shepherd to limit the spread of sheep and to protect them from predators.

The data were collected during three consecutive days. All measurements were always collected at 7:00 to 8:30 a.m. in each site for each sheep separately. The assessment of well-being was based on the measurement of basic physiological parameters, allowing us to determine the physiological adaptation of the body to environmental conditions, such as heart rate, respiratory rate, rectal temperature, body condition, and body weight. Rectal body temperature was measured by inserting to the rectum an electronic thermometer to a depth of 9 cm. The heart rate was examined with a stethoscope, while the number of breaths was examined by observing the chest, abdominal movements, and auscultation with a stethoscope. The visual estimate of the body score condition was performed according to Russel et al. [16], ranging from 2.0 to 4.5. All ratings of the body condition score were performed by the same observer. Blood samples were collected from the jugular vein of ewes 2 weeks before the grazing period in sheepfold (Bezek) and in every grazing site after the animals had stayed there for 3 weeks: Gródek (mid-July), Kąty (end of August), and Staw (mid-October). These days air temperature was recorded and mean temperature was calculated (8:00 a.m.; 4:00 p.m.; midnight). The animals were weighed before the beginning of the grazing period (in May in the sheepfold) and after the grazing period (in October).

The analyses were conducted in the laboratory. We performed haematology using an automated haematological analyser, Abacus Junior Vet (Diatron, Hungary), with species-specific software using the impedance method. Plasma for analysis of the biochemical parameters and cholesterol was obtained by centrifugation of whole blood at 3000 rpm ($603\times g$) for 15 min in a laboratory centrifuge (MPW-350R; MPW Medical Instruments, Warsaw, Poland) at 4 °C. The cortisol concentration was determined with an enzyme immunoassay kit (Immulite 2000 Cortisol, Siemens, UK) according to the manufacturer's instructions. Total cholesterol and glucose analyses were performed using the automated spectrophotometry method according to the recommended protocol (Chemical Autoanalyzer BS-120, Mindray, Shenzhen, China).

*2.3. Vegetation Field Study*

Studies concerning the vegetation of xerothermic dry grassland have been conducted since 2008. This paper presents the data collected during the growing season of 2018. A total of 36 phytosociological relevés from 3 sites (12 from each site: Staw, Gródek, Kąty) were used for the vegetation analysis. The relevés were collected using the Braun–Blanquet method [17] on an area of 25 m$^2$ in homogeneous patches representing plant communities. The vascular plant nomenclature according to Mirek et al. [18] and the classification of communities according to Matuszkiewicz et al. [19] and Mucina et al. [20] were used. Based on the cover-abundance of species and ecological indicator values by Ellenberg et al. [21], climatic and edaphic conditions were assessed. On this basis, the analysed areas were characterised by diverse habitat conditions.

### 2.4. Grassland Assesment

The grassland quality index (EGQ) was calculated based on the average cover by individual plant species and their fodder values determined by Novak [22]. The abundance of each species was recorded on the mean cover-abundance scale transformed as follows: r = 0.1%, + = 0.5%, 1 = 2.5%, 2 = 15%, 3 = 37.5%, 4 = 62.5%, and 5 = 87.5%. For calculations, the obtained values were transformed proportionally so that the sum of all species was 100%. The climatic (L—light, T—temperature, K—continentality) and edaphic (F—moisture, R—pH, soil reaction, N—nitrogen content) conditions were assessed using the cover-abundance of species and ecological indicator values by Ellenberg et al. [21].

### 2.5. Statistical Analyses

Statistical procedures were performed using Statistica version 13.1 software (Statsoft Poland). We used a two-way analysis of variance (ANOVA) for repeated measurements to assess the effects of sampling time and their interaction on the studied parameters in ewes. When the effect of a factor was significant ($p < 0.05$), we performed a post hoc Tukey test to determine specific differences between the means. The *Student's t-test was used* for comparing data between breeds.

## 3. Results

Phytosociological studies carried out in three Natura 2000 sites confirmed the presence of valuable natural habitats. In Habitat 6210-3, flowering xerothermic grasslands from the *Cirsio-Brachypodion pinnati* alliance (Festuco-Brometea class) predominated at that site. In the area of Stawska Góra (Staw), communities of calcareous grassland (All. *Cirsio-Brachypodion pinnati*) occurred in a mosaic with patches of thickets from the *Rhamno-Prunetea* class. In Gródek, a *Thalictro-Salvietum pratensis* association and a community with *Brachypodium pinnatum* were found. Kąty was a priority habitat with significant Orchidaceae sites, with a predominant association of *Inuletum ensifoliae* and an initial community with the predominance of *Anthyllis vulneraria*. Fallow land was covered by a community with *Elymus repens* corresponding to the *Convolvulo arvensis–Agropyretum repentis* association (Table 1). Thickets of the Rhamno-Prunetea class, which occurred mainly in Staw (in the form of smaller patches, also in Kąty), are a shelter for grazing sheep, but at the same time, they pose a great threat to xerothermic grasslands.

The evaluation of grassland quality (EGQ) according to Novák et al. [22] showed that the value of biomass from the analysed grasslands was very low. The biomass values most often ranged between less and least valuable (Figure 1). This demonstrates the difficult nutritional conditions for grazing sheep.

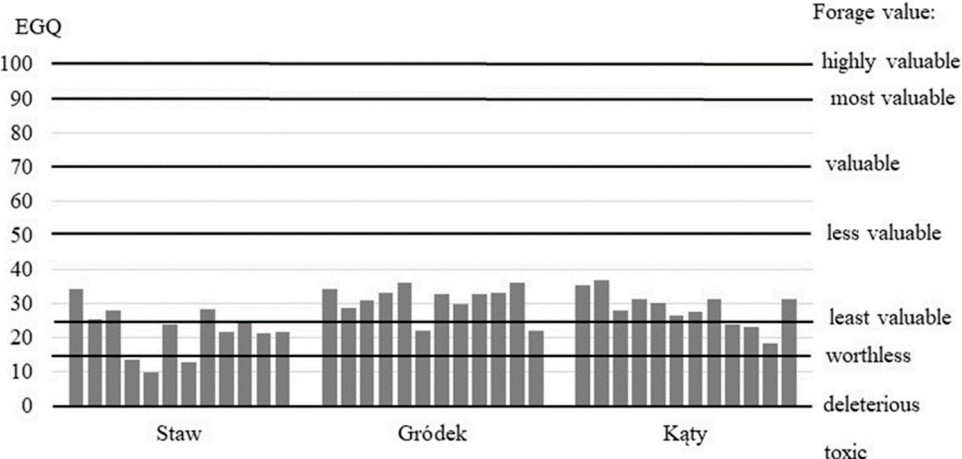

**Figure 1.** Evaluation of grassland quality (EGQ).

The analysed areas were characterised by diverse habitat conditions (Figure 2) determined on the basis of plant species and ecological indicators by Ellenberg et al. [21]. The greatest diversity was observed in Staw due to the mosaic of communities of xerothermic grasslands and thermophilic shrubs.

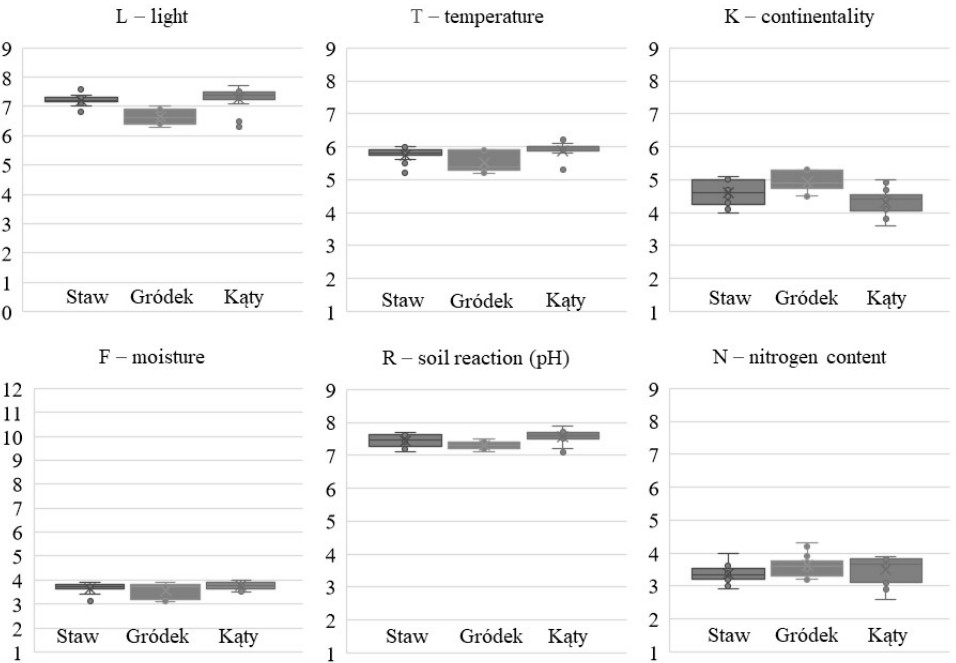

**Figure 2.** Climatic and edaphic conditions based on ecological indices.

Based on ecological indicator numbers, the following scales were used: from 1 (shadiest habitats) to 9 (full light) for light (L), from 1 (coldest areas) to 9 (warmest areas) for temperature, from 1 (euoceanic) to 9 (eucontinental) for continentality, from 1 (very dry soils) to 12 (water) for soil moisture, from 1 (very acidic) to 9 (alkaline) for soil reaction pH, and from 1 (poorest soils) to 9 (very fertile soils) for nitrogen content.

Based on ecological indicator numbers, on a scale of 1 (shadiest habitats) to 9 (full light) for light (L), 1 (coldest areas) to 9 (warmest areas) for temperature, 1 (euoceanic) to 9 (eucontinental) for continentality, from 1 (very dry soils) to 12 (water) for soil moisture, from 1 (very acidic) to 9 (alkaline) for soil reaction pH, from 1 (poorest soils) to 9 (very fertile soils) for nitrogen content

Climatic conditions were defined as partial shade to full light with periodic shading (light index), thermal conditions between moderately cool and moderately warm (temperature index), and climate between suboceanic and slightly subcontinental (continentality index). Based on edaphic conditions, these habitats were dry (moisture index), slightly acidic, or neutral (pH index) and located on nitrogen-poor soils (nitrogen content index). This demonstrates the very difficult climatic and edaphic conditions in the context of sheep grazing.

At the start of grazing, Uhruska sheep were significantly better in condition and body weight compared to Świniarka sheep ($p < 0.05$). During grazing, the Świniarka sheep improved their condition, significantly increasing their average body weight ($p < 0.05$). However, the condition and body weight of the Uhruska sheep did not change significantly (Table 2).

**Table 2.** Physiological parameters in ewes (*n* = 40).

| Sites Temperature (°C) | Breed | Bezek (Sheepfold) (22 °C) | Gródek (23 °C) | Kąty (28 °C) | Staw (12 °C) | B | S | BW | Interaction |
|---|---|---|---|---|---|---|---|---|---|
| | | **Nature 2000 Site/Place** | | | | **Effect of Factors** | | | |
| Body condition (scale 1–5) | UHR | 4.5 ± 0.4 * | 4.1 ± 0.4 | 3.9 ± 0.5 | 4.0 ± 0.5 | * | * | * | BxS |
| | SWI | 2.1 ± 0.6 * | 3.5 ± 0.3 | 3.4 ± 0.6 | 3.9 ± 0.5 | | | | |
| Body weight (kg) | UHR | 73.1 ± 10.5 * | - | - | 68.7 ± 8.1 * | - | - | - | - |
| | SWI | 29.5 ± 3.7 a, * | - | - | 37.3 ± 6.0 b, * | | | | |
| Rectal temperature (°C) | UHR | 39.9 ± 0.3 b | 39.2 ± 0.2 a | 39.3 ± 0.2 a, * | 39.3 ± 0.3 a | * | * | NS | BxS |
| | SWI | 39.7 ± 0.2 c | 39.4 ± 0.3 b | 38.9 ± 0.3 a, * | 39.3 ± 0.2 b | | | | |
| Respiration rate (/min) | UHR | 92.6 b ± 24.5 b, * | 59.4 ± 13.5 a | 86.0 ± 23.1 b | 74.4 ± 20.4 a, b | * | * | NS | BxS |
| | SWI | 50.2 ± 12.4 a * | 57.6 ± 16.2 a | 82.4 ± 20.8 b | 64.8 ± 15.8 a, b | | | | |
| Heart rate (/min) | UHR | 146.0 ± 25.0 | 145.4 ± 32.1 | 135.6 ± 20.9 | 137.2 ± 38.6 | * | NS | NS | |
| | SWI | 135.6 ± 22.9 | 130.4 ± 35.3 | 122.8 ± 27.1 | 120.4 ± 27.3 | | | | |

Values represent the mean ± standard error; means within each row with different letters are significantly different (*p* < 0.05). Means within Uhruska and Świniarka breeds (UHR/SWI) denoted with * are significantly different (*p* < 0.05) between Uhruska and Świniarka breed; B—breed, S—site; BW—body weight; BxS interaction: breed x site; NS, non-significant; * (*p* < 0.05).

During observation, a significantly higher rectal temperature was recorded in the sheepfold compared to the sites during grazing (*p* < 0.05). During grazing in the naturally valuable areas of the Lublin region, respiration rates varied depending on where the animals stayed. The Uhruska breed was characterised by a higher respiration rate at all grazing sites compared to the Świniarka breed, but a statistically significant difference was found only in the sheepfold (*p* < 0.05). The higher ambient temperature did not increase the heart rate.

The level of leukocytes (WBC) in particular periods and grazing positions did not change; however, a difference between breeds was observed (Table 3). A significantly higher mean level was observed in the Świniarka compared to the Uhruska breed (*p* < 0.05). In both breeds, there was no influence of the place of grazing on the number of leukocytes, but in the Uhruska breed, significant differentiation was observed in the individual fractions of lymphocytes (LY%) and granulocytes (GR%). During the grazing of the animals in Staw, the highest LY% and the lowest GR% were observed in comparison to other places where the animals were staying.

The influence of the grazing site on some red blood cell parameters was noted during the observations. For the Świniarka breed, the lowest level of erythrocytes (RBC) and haemoglobin (HGB) was recorded when the sheep stayed in the sheepfold before the beginning of the pasture season (*p* < 0.05). Differences in red cell indices were also observed between breeds. Significantly higher amounts of RBC and higher levels of HGB and haematocrit (HCT) in the blood were found in Uhruska sheep (*p* < 0.05). However, during grazing, a dynamic increase in mean haemoglobin content (MCH) and mean blood cell volume (MCV) was observed in the Świniarka breed.

The lowest levels of cholesterol and glucose in the blood of sheep in both breeds were observed in the sheepfold (Table 4). At individual sites during grazing, a statistically significant increase in these indices in the blood was observed in comparison to the sheepfold and this trend was maintained until the end of grazing (*p* < 0.05). No statistically significant changes were observed in the blood cortisol levels.

**Table 3.** Haematological parameters in the blood plasma of ewes (*n* = 40).

| Sites | Breed | Nature 2000 Site/Place | | | | Effect of Factors | | | Interaction |
|---|---|---|---|---|---|---|---|---|---|
| | | Bezek (Sheepfold) | Gródek | Kąty | Staw | B | S | BW | |
| WBC ($10^3$/μL) | UHR | 6.2 ± 2.0 * | 6.7 ± 2.4 * | 6.3 ± 1.9 * | 6.7 ± 1.9 * | * | NS | NS | |
| | SWI | 10.8 ± 3.8 * | 11.1 ± 4.0 * | 10.8 ± 3.0 * | 10.9 ± 4.0 * | | | | |
| LY% | UHR | 57.2 ± 6.7 a, b | 54.6 ± 6.8 a * | 57.2 ± 9.9 a, b | 64.5 ± 6.9 b * | * | * | NS | |
| | SWI | 51.1 ± 9.6 | 42.4 ± 12.3 * | 50.9 ± 9.6 | 47.6 ± 12.0 * | | | | |
| GR% | UHR | 42.3 ± 6.7 a, b | 44.8 ± 6.8 b * | 42.3 ± 9.9 a, b | 35.0 ± 6.9 a * | * | * | NS | |
| | SWI | 48.4 ± 9.6 | 57.1 ± 12.3 * | 48.5 ± 9.6 | 50.4 ± 11.8 * | | | | |
| RBC ($10^6$/μL) | UHR | 9.5 ± 6.7 * | 9.6 ± 6.8 | 9.7 ± 9.9 * | 9.7 ± 6.8 | NS | NS | * | |
| | SWI | 7.5 ± 9.6 a * | 9.2 ± 12.3 b | 7.9 ± 9.6 a, b, * | 8.4 ± 11.8 a, b | | | | |
| HGB (g/dL) | UHR | 12.0 ± 1.5 * | 13.1 ± 1.1 | 12.3 ± 0.9 * | 12.3 ± 1.3 | NS | * | * | |
| | SWI | 9.7 ± 1.2 a * | 11.7 ± 1.5 b | 10.4 ± 1.9 a, b * | 11.8 ± 2.5 b | | | | |
| HCT (%) | UHR | 28.6 ± 3.4 * | 29.5 ± 2.6 | 28.0 ± 2.2 * | 27.7 ± 3.0 | * | * | * | |
| | SWI | 24.7 ± 2.6 a, b * | 26.7 ± 3.5 a, b | 24.3 ± 4.4 a * | 27.9 ± 5.4 b | | | | |
| MCV (fl) | UHR | 30.0 ± 1.7 | 28.2 ± 1.4 | 28.9 ± 1.8 | 28.8 ± 2.1 * | * | * | NS | |
| | SWI | 33.1 ± 3.0 a, b | 29.4 ± 2.9 a | 31.4 ± 4.8 a | 35.6 ± 10.1 b, * | | | | |
| MCH (pg) | UHR | 12.6 ± 0.5 | 12.6 ± 0.6 | 12.7 ± 0.5 | 12.7 ± 0.6 * | * | * | NS | BxS |
| | SWI | 13.1 ± 0.8 a | 12.9 ± 0.7 a | 13.4 ± 1.7 a, b | 14.8 ± 3.4 b, * | | | | |

Values represent the mean ± standard error; means within each row with different letters are significantly different (*p* < 0.05); differences between Uhruska and Świniarka breeds (UHR/SWI) denoted with * are significant (*p* < 0.05) between Uhruska and Świniarka breed. Abbreviations: WBC, white blood cells; LY%, percentage lymphocytes; GR%, percentage granulocytes; RBC, red blood cells; HGB, haemoglobin; MCV, mean corpuscular volume; HCT, haematocrit; MCH, mean corpuscular haemoglobin; B—breed; S—site; BW—body weight; BxS interaction: breed x site; NS, non-significant; * (*p* < 0.05).

**Table 4.** Biochemical parameters in the blood plasma of ewes (*n* = 40).

| Sites | Breed | Nature 2000 Site/Place | | | | Effect of Factors | | | Interaction |
|---|---|---|---|---|---|---|---|---|---|
| | | Bezek (Sheepfold) | Gródek | Kąty | Staw | B | S | BW | |
| cholesterol (mg/dL) | UHR | 35.2 ± 13.2 a, * | 54.1 ± 13.1 b | 59.5 ± 9.8 b | 51.7 ± 13.8 a, b | NS | * | NS | |
| | SWI | 34.2 ± 4.6 a, * | 60.1 ± 12.5 b | 49.6 ± 14.6 a, b | 42.2 ± 10.4 a, b | | | | |
| cortisol (μg/dL) | UHR | 2.2 ± 1.2 | 2.7 ± 0.9 | 2.7 ± 1.4 | 1.4 ± 0.64 | NS | NS | NS | |
| | SWI | 2.8 ± 1.4 | 3.0 ± 1.0 | 2.2 ± 1.2 | 1.9 ± 1.4 | | | | |
| glucose (mmol/L) | UHR | 3.1 ± 0.4 a | 3.8 ± 0.3 b | 3.6 ± 0.6 b | 3.6 ± 0.4 b, * | * | * | NS | BxS |
| | SWI | 2.7 ± 0.6 a | 3.7 ± 0.3 b | 3.7 ± 0.8 b | 3.0 ± 0.5 a, * | | | | |

Values represent the mean ± standard error. Means within each row with different letters are significantly different (*p* < 0.05). Differences between Uhruska and Świniarka breeds (UHR/SWI) denoted with * are significant (*p* < 0.05) between Uhruska and Świniarka breed; B—breed, S—site, BW—body weight; BxS interaction: breed x site; NS, non-significant; * (*p* < 0.05).

## 4. Discussion

The analysed research areas (Staw, Gródek, and Kąty) are protected due to their important function in preserving biodiversity. They are habitats of many rare plants and animals, including rare species threatened by extinction [3,4,22]. Grazing small ruminants in xerothermic grasslands is the best method of protection [4,8,23].

When looking at the nutritional needs of sheep, it should be noted that the vegetation of xerothermic grasslands is a poor fodder for animals [5,8,23], which was confirmed by

the conducted research (Table 1, Figure 1). However, some diversity of vegetation was observed, which could have influenced the conditions and parameters of grazing sheep. Based on Novak's [22] numbers, sheep in Gródek and Kąty received slightly better feed, while the worst feed was received in Staw. Most grasses (mainly *Elymus hispidus* subsp. *hispidus* and *Brachypodium pinnatum*) and legumes with higher yielding potential grew on the dry grasslands in Gródek. In turn, Staw had the most thermophilic thickets that could provide shelter for grazing sheep. On the other hand, there are also poisonous plants, e.g., *Adonis vernalis* growing on xerothermic grassland. To estimate the direction and intensity of physiological changes, the physiological and haematological indicators of sheep's blood were analysed in various places of grazing.

Heat stress may be a factor limiting the productivity of animals and disturbing their welfare. In the conditions of pasture grazing in summer, the reason is high temperatures. The consequence is reduced feed intake, increased water intake, and slower metabolism in animals. This could negatively affect growth, productivity, reproduction, feed utilisation, and milk production, causing significant economic losses [24]. The control of body weight and the condition of animals on a point scale can be a helpful tool in the proper management of feed and the decision for possible supplementation during grazing under extensive conditions [25].

At the beginning of grazing, better conditions and body weights were found in Uhruska sheep compared to Świniarka sheep. Uhruska sheep are a general utility breed with good prolific and meat parameters. The Świniarka breed is a primitive type. During grazing, Świniarka sheep improved their condition, significantly increasing their average body weight. However, data on the condition and body weight of Uhruska sheep showed a decreasing trend, although without statistically significant differences. This may indicate better use of poor fodder during grazing by Świniarka sheep.

In the experiment, the animals had constant and unlimited access to water. According to Jaber et al. [26], limiting the availability of water may contribute to weight loss because of a decrease in food intake. In the cited studies, giving water every 4 days for a period of 6 weeks resulted in a 5% reduction in the body weight of animals, while maintaining thermal stability. Restricting water availability for 2 days resulted in minimal weight loss.

One of the parameters that allows us to determine the body's response to heat stress is the measurement of rectal temperature. Significantly higher rectal temperatures were recorded in both breeds during the animals' stay in the sheepfold ($p < 0.05$). An important piece of information obtained during the observation was the balanced rectal temperature in the animals during grazing, which was within reference standards and balanced in both breeds of sheep [27]. This is especially important in the context of research by McManus et al. [27], who showed that when the ambient temperature reached 32 °C in animals, the rectal temperature rose above normal. However, an increase in rectal temperature of 1 °C reduces production parameters in most farm animals.

During grazing in the naturally valuable areas, the respiration rate, indicating high thermal stress, was not observed,. However, differences in the respiration rate, depending on the location of the animals, were noted. This could be related to the different ambient temperatures occurring in individual sites of grazing. In a high-stress situation during thermal stress, the respiratory rate is 80–120 breaths/per minute, while in a low-stress situation, the respiratory rate is much lower—40–60 breaths/per minute [28]. The increase in respiratory rate is caused by the direct stimulation of peripheral temperature receptors that transmit nerve impulses to the hypothalamus, among other causes [28]. In individuals in the sun in Mediterranean regions, the respiratory rate is up to 50% higher compared to individuals in the shade. Similar trends were observed in sheep kept in Brazil, where the respiratory rate at lower temperatures was lower than that recorded at higher temperatures [29]. Monitoring of the heart rate from the technical site is very complicated. Measurements made using adopted wearing devices such as a polar tester could have been more reliable [30].

During observations in the sheepfold, a large variation in respiratory rate between breeds was observed. In Uhruska sheep, a significantly higher respiratory rate was recorded in the sheepfold than in the pasture in Gródek, which may be due to poorer ventilation inside the building. That is very interesting and there is still a question that these differences were not observed in Świniarka sheep. According to Sevi et al. [31], a simple and effective way to protect animals against the effects of thermal stress is to protect the presence of shaded places in grazing areas. The ability to adapt particular sheep breeds to grazing is evidenced by the measurement of heart rate. In both breeds, the ambient temperature during grazing did not significantly change the heart rate. This could indicate that both breeds were well adapted to environment.

In veterinary practice, blood tests are of great importance to help diagnose not only many diseases, but also to ensure animal welfare. Poor housing and nutrition conditions may result in health disorders and may exceed the reference standards of haematological and physiological parameters. The level of individual blood elements depends on many factors, including species, breed, housing conditions, production, nutrition, and physiological conditions [32].

Haematological examinations usually include the determination of the number of leukocytes, erythrocytes, and platelets, determination of the white-cell image, haemoglobin concentration, and red cell indices, which reveal the assessment of the overall homeostasis of the body. When animals are subjected to heat stress, the hypothalamic/pituitary/adrenal axis is activated, and glucocorticoids are released from the adrenal cortex. Therefore, in stressful situations, an increase in cortisol, hyperglycaemia, leucocytosis with neutrophilia, lymphopenia, and eosinopenia are observed [33]. Therefore, we analysed white blood cells, percentage lymphocytes, percentage granulocytes, cortisol, and glucose in blood in both breeds of sheep. In the observations carried out during grazing in the xerothermic grasslands of the Lublin region, the analysed haematological indices were within the reference values and therefore did not affect the animal welfare (Table 3). There was a significant interracial difference in leukocyte (WBC) levels. A higher mean WBC count was observed in the Świniarka breed compared to the Uhruska breed, as well as significant differences in lymphocyte (LYM%) and granulocyte (GRA%) fractions. Notably, there is a direct relationship between leukocyte profile, stress, and glucocorticoid levels. Glucocorticoids, such as cortisol, whose increased secretion occurs in stress, influence a reduction in lymphocytes and an increase in the number of neutrophils, and neutrophils are the main fraction of granulocytes [33]. The relationship between granulocytes/lymphocytes can be used as a physiological indicator of stress. During the animals' stay in Gródek, the lowest % share of LY and the highest % share of GR were recorded. At the same time, high levels of cortisol were also noted in both breeds, although there were no statistically significant differences. These physiological changes could be related to stress and the release of cortisol. Although the research shows (Figure 2) that the pasture in Gródek had the best nutritional conditions, the observed changes in blood indices were caused by the change in the animals' place of residence from the sheepfold to the pasture in Gródek. Blood cortisol levels tended to decrease as they were at their lowest at the end of grazing.

Higher values of red blood cell indices were found in Uhruska animals. Grazing had a positive effect on blood indices in the Świniarka breed as they significantly increased RBC, HGB, and MCH. Finally, at the end of grazing, RBCs in Świniarka were characterised by a higher haemoglobin content in the blood cells (MCH, MCV). The elevated Hb concentration during grazing compared to the sheepfold in the Świniarka breed could be attributed to the fact that these animals need more Hb to carry oxygen molecules to support the evaporative cooling mechanism.

Many studies conducted under various environmental conditions have confirmed the impact of limited water availability on haematocrit levels [34–36]. The reduced value of the haematocrit level in Świniarka breed can be interpreted as an indicator of dehydration or anaemia during grazing in the Kąty region. This significant reduction in HCT blood levels in Świniarka breed in Kąty could be attributed to dehydration because of both heat stress

and increased respiratory evaporative cooling and sweating mechanisms because of higher air temperature during summer season. There is still a question that these differences were not observed in Uhruska sheep. Providing shade would prevent the deterioration of the immune system, which is observed when animals stay at high temperatures and thus reduces susceptibility to diseases and pathogens [34]. During grazing in Gródek, Kąty, and Staw, sheep were provided shade in bushes and the shed (Photo 1, 2, 3).

The high adaptability of sheep to grazing in xerothermic grasslands is evidenced by the determined level of glucose in the blood. In all analysed positions, it was within the reference standards for both breeds (2.8–4.5 mmol/L) [37]. Higher glucose levels were observed in animals under thermal stress. In local breeds better adapted to extreme temperatures under the influence of heat stress, an increase in blood glucose levels can be observed, as observed in Uhruska and Świniarka sheep during grazing in Gródek and Kąty (but never over standards). This is probably due to the mobilisation of fat reserves that occurred in response to the increasing demand for energy, which is associated, for example, with a higher respiratory rate and lower feed intake under these conditions [38].

The association between stress and increased secretion of cortisol in small ruminants is well documented [39,40]. In the current research, no statistically significant changes were observed in the blood cortisol levels, although fluctuations can be observed. These fluctuations could be affected by season [41]. According to Seijan et al. [39], the highest cortisol level was shown in sheep under thermal and combined stress (thermal and nutritional). In the nutritional stress group, the lowest level of cortisol was observed. It is obvious that manipulations elevated plasma cortisol levels, but all animals were exposed to that stress and this factor affected all animals equally [42].

## 5. Conclusions

Sheep grazing was carried out on xerothermic grasslands, which are habitats with a high nature value. However, the analysed grasslands were characterised by very difficult climatic and edaphic conditions (based on Ellenberg's indices) [21] and low fodder values (based on Novak's evaluation of grassland quality) [22] for sheep.

As a result of the conducted analyses, grazing in environmentally valuable areas and sometimes with poor fodder did not adversely affect the welfare of grazed sheep. In the Uhruska and Świniarka breeds, there were no indicators demonstrating the presence of heat or nutritional stress, but physiological changes were observed as a result of the animals' adaptation to new environmental conditions. Despite poor fodder, the Świniarka sheep improved their condition and body weight. Fluctuations in parameters indicating an adaptation process were more visible in this breed. These findings further strengthen the fact that this breed possesses an adaptive capability superior to the Uhruska breed.

**Author Contributions:** Designed the study: M.G.-K. and M.K. Organised sample collection: M.G.-K., P.N. and K.P. Performed the research: M.G.-K. and M.K. Data acquisition: A.H.-S., K.P. and A.J. Analysed the data: M.G.-K. and M.K. Performed statistical analyses: M.G.-K. and A.H.-S. Drafted the manuscript and submitted the article for publication: M.G.-K. and P.N. Involved in the discussion of the manuscript: M.G.-K., M.K. and A.J. All authors have read and agreed to the published version of the manuscript.

**Funding:** This research was supported by the "Uses and conservation of farm animal genetic resources under sustainable development" project co-financed by the National Centre for Research and Development in Poland within the framework of the strategic R&D programme "Environment, agriculture and forestry" BIOSTRATEG, contract number: BIOSTRATEG2/297267/14/NCBR/2016.

**Institutional Review Board Statement:** Not applicable.

**Informed Consent Statement:** Not applicable.

**Data Availability Statement:** The data presented in this study are available on request from the corresponding author. The datasets used and analysed during the current study are available from the corresponding author on reasonable request.

**Conflicts of Interest:** The authors declare no conflict of interest.

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
