# Peer review of "Some Physiological Responses of Native Sheep Breeds to Environmental Conditions during Grazing in Natura 2000 Habitats"

_agriculture, doi:10.3390/agriculture13050982_

Round 1

Reviewer 1 Report

The Review of the article 'Assessment of the response of native sheep breeds to environmental conditions during grazing in Natura 2000 habitats'

The article aims to evaluate the welfare of native sheep breeds under challenging environmental conditions and to confirm the hypothesis that native sheep breeds can be used to protect natural habitats.

In my opinion, this article does not make a significant contribution to science. In addition, the article contains some misleading statements.

Major items

Repeated Measures ANOVA is an appropriate method for comparing the results of each breed between samplings. However, I do not believe it is an appropriate method for comparing data between breeds.

The statistical analysis does not include correlation between environmental factors and the results of hematological, biochemical, or physiological parameters. All these data are shown in the Results section, and the correlation coefficient can be easily calculated. For example, in Lines 264 - 271, the authors speculate that the frequency of respiration depends on the ambient temperature. The correlation test can confirm or deny these speculations. The same is true for cortisol levels (Lines 305 - 310).

 In addition, blood cortisol levels are influenced by circadian and seasonal rhythms and, of course, by stressful situations. To estimate cortisol status, samples should be collected at the same hour each time. No effects of season on cortisol levels were mentioned in the manuscript, although fluctuations can be observed. In addition, it was described several times that manipulation with the animals and blood collection can cause stress and increase cortisol levels. This should be noted and discussed.

Table 2 indicates that body condition was estimated on a scale of 1 to 5. The body condition of the UHR breed in sheepfold was estimated to be 5.5. How is this possible?

Lines 319 - 329. A decrease in hematocrit is not an indicator of dehydration. It is not clear how the authors explain the changes in hematocrit.

Based on the comments, an improved version of the Discussion should be created.

Minor items

Material and Methods does not describe how the data for light, temperature, humidity, etc. were obtained. I wonder if the measurements were always taken at the same hour?

Figure 2. Although the reference is given (Ellenberg et al.), it would be much easier for the reader if the indices for all parameters were explained. How were they calculated? What do the values given mean?

Line 260. The Lublin region is not mentioned in the tables.

Author Response

Dear Reviewer

Please find enclosed the manuscript entitled “Assessment of the response of native sheep breeds to environ-mental conditions during grazing in Natura 2000 habitats” by Monika GreguÅ‚a-Kania et al., which was corrected according final suggestions .

Thank you for valuable comments and suggestions. All comments were taken into account and we have improved our manuscript:

Reviewer 2 Report

Dear Authors

Thank you for this nice study on an important subject.

There are some aspects in the methods part and in the result part which are not yet clear to me (see remarks below). I ask you to ameliorate these sections of your work.

Line 22: replace “that could indicate the presence of” by “for”

Line 25: replace “indicating” by “showing”

Line 26: please explain: more visible than what?

Line 48: is it the nutritional value of milk for human consumption or for the lambs? Was this a subject in this trial? Milk is never mentioned later on. Please specify why you use this reference here, why this aspect is important.

Line 54: I don’t understand why “reduced space” and “isolation” could be a stress factor in extensive systems, because normally animals are kept outside in large areas when they are kept extensively. Please explain closer how you define “extensive production technologies” and why those two factors cause stress in these systems.

Line 63: replace the second last word on this line “to” by “living in”

Lines 69 – 74: Here you don’t use the same order of mentioning the three sites as below (in chapter 2.2.). It would make it easier for the reader if always the same order would be used.

Line 77: were there 40 ewes of each breed or 40 all together? Were there no male animals and no lambs? Please mention that.

Line 81: What is an indoor pasture system? I have never heard of that and couldn’t find anything about it in the internet. Pasture systems are usually outdoor systems. Please explain.

Line 83: In which year were those trials carried out? Please add the year after “October”

Line 84: was there no fencing within the pastures? Please add if there was or not and why.

Line 90: Was temperature measured in the rectum? Please specify.

Line 96: Were the observations carried out the same day as blood sampling? (please specify). “assessment of individual conditions”: what does it mean? Please specify exactly what you observed (bcs, weight, body temperature?). Did you observe every single ewe? (please specify).

Line 104: Was cortisol determined in plasma? (please specify)

Line 112: Please add in the brackets: (12 from each site: Staw, Grodek, Katy)

Lines 118 – 125: please make a separate chapter for 2.3. “Grassland assessment”

Line 121: please replace the “-“ by “=”; that way it will be much easier to read.

Lines 126 – 130: please use this paragraph for chapter 2.4. “Statistical analyses”

Figure 1: maybe you could again mention that the values refer to nutritional quality (either in the subtitle or above the description of the different qualities)

Lines 152 – 154: This paragraph belongs to the methods part (until the brackets on line 154). Please remove it and place it in the methods part an d write a sentence here to introduce figure 2.

Figure 2: It is not clear what the scores in those graphs mean. Please explain them all in the subtitle. (It is not enough to just give a reference here, because it is a very important graph, which should be understandable without consulting the reference).

Line 172: delete “rectal temperature was recorded in the animals.” and continue the sentence with “a significantly….”

Line 181: delete “within Uhruska and Swiniarka breed (UHR/SWI)”

Line 182: add after (P<0.05): “between Uhruska and Swiniarka breed”.  (because there is a difference between breeds and not within one breed!)

Line 201: Replace “Means within a” by “Differences between”

Line 202 replace “significantly different” by “significant”

Line 212: Replace “Means within a” by “Differences between”

Line 213 replace “significantly different” by “significant”

Line 227: is Adonis really poisonous for sheep? Is it really true that Adonis vernalis grows in thickets? As I saw it up to now they grow mainly in open areas. Please do some research  whether this is really true.

Line 232: delete “the” before summer

Line 235: here you are talking about milk production again, but was this a subject in your study? Were there any lambs? Please specify.

Line 245: add “Swinjarka” before sheep.

Line 247: It is not clear why unlimited access to water should lead to weight loss. Please explain.

Lines 260 – 264: Those are results of your study you hadn’t mentioned in the results part. Please move them to the results part. You should not discuss results here that you hadn’t mentioned in the results part!

Line 262: number of breaths “per minute”(please add “per minute”, if it is the case)

Line 266: Please add a reference to this statement.

Line 273/274: replace “frequency2 by “rate”

Line 274: add “in Grodek” after pasture (because it was only significant in Grodek)

Line 275: You write that this might be because poorer ventilation inside the building. But why did this not affect the Swiniarka sheep? Please provide an explanation for that or remark that there is still a question.

Line 279: Would that mean that both breeds were well adapted? If so, please mention it here.

Line 285: Maybe “production” should be added here as a factor. Please give a reference for that statement.

Line 292: Please add a sentence here: “Therefore we analysed….” And mention again what you did in reference to that statement.

Line 295: replace “the reduction of” by “animal”

Line 298: Do you have an explanation for that?

Line 322: Please replace that line by: “The reduced value of the haematocrit level can be interpreted as an indicator of dehydration…

Line 323: add “because it” after the comma

Line 324: Why was this not the case in Uhruska breed? Please explain or at remark at least that it was not the case.

Line 327: replace “prevents” by “would prevent”

Line 329: Please add whether there was shade in you experiment; in which sites?

Line 333: replace “noted” by “observed”

Line 335: add in brackets: “(but never over standards”)

Line 336: replace “is” by “was” and “occurs” by “occurred”

Line 347: replace “in” by “as a result of”

Line 249: replace “indicators” by “parameters”

Line 350: Add “in this breed” after visible.

Author Response

(The authors gave the same response as above.)

Reviewer 3 Report

There are not many breed comparisons like this, and even fewer that are genuinely multidisciplinary, so this study is very welcome. There are a few relatively minor problems. See attached document.

Certain aspects of the study need to have more emphasis. There needs to be an adequate description of the two breeds involved, and some evidence that the authors actually know something about animal welfare science, and not just about veterinary medicine. A fundamental part of animal welfare science is that welfare (which is correctly defined as "the extent to which an animal is coping with its environment") should be assessed by a variety of measures - for example, there should be behaviour observations as well as physiological. There should also be a proper account of how the study actually proceeded. 

Author Response

(The authors gave the same response as above.)

Reviewer 4 Report

Reviewer' comments for the Authors;

 The manuscript is important in terms of providing valuable information about “Assessment of the response of native sheep breeds to environmental conditions during grazing in Natura 2000 habitats” It is well written in English. There is no need for linguistic revision. I make some important recommendations for improving the proposed paper. 

1. The work is very nice as an idea. I really liked the subject. A study that can inform us about the use of pastures in terms of sustainability. But there are serious errors in experimental design. According to the authors' experiment plans, the sheep used each pasture for 7 weeks (21 weeks total). That is, from the beginning of June to the end of October. Do the changing vegetation and climatic conditions in this process have an effect on the results obtained from the animals? I could not find answers to these questions in the article.

2. In addition, changing temperatures and the data obtained (for example heat stress or blood parameters) can be correlated and presented in graphs. It is not possible to draw clear conclusions from the tables where only the averages are given.

3.  Have you studied interactions (such as breed x region)? I think they should be given.

4. The initial weights of the two breeds are quite different from each other. In order to say whether it performed well or badly during the study, the weights per trial can be put into the model as a covariate.

5. You can talk to a statistician and get ideas from him in the analysis and presentation of the values ​​you get.

6.   Some other corrections and explanations have been made on the article.

Author Response

Dear Reviewer

Please find enclosed the manuscript entitled “Assessment of the response of native sheep breeds to environ-mental conditions during grazing in Natura 2000 habitats” by Monika GreguÅ‚a-Kania et al., which was corrected according final suggestions .

Thank you for valuable comments and suggestions. All comments were taken into account and we have improved our manuscript

Thank you for your consideration

Yours faithfully,

Round 2

Reviewer 4 Report

When the article was examined, the authors made the changes requested by the reviewers in the article. Only the article has some minor typos. These need to be revised and corrected.